# Development of GaN HEMTs Fabricated on Silicon, Silicon-on-Insulator, and Engineered Substrates and the Heterogeneous Integration

**DOI:** 10.3390/mi12101159

**Published:** 2021-09-27

**Authors:** Lung-Hsing Hsu, Yung-Yu Lai, Po-Tsung Tu, Catherine Langpoklakpam, Ya-Ting Chang, Yu-Wen Huang, Wen-Chung Lee, An-Jye Tzou, Yuh-Jen Cheng, Chun-Hsiung Lin, Hao-Chung Kuo, Edward Yi Chang

**Affiliations:** 1Department of Photonics and Institute of Electro-Optical Engineering, College of Electrical and Computer Engineering, National Chiao Tung University, Hsinchu 30010, Taiwan; alger.99g@g2.nctu.edu.tw (L.-H.H.); itriA30378@itri.org.tw (P.-T.T.); cath01.ee09@nycu.edu.tw (C.L.); s922085493@gmail.com (Y.-T.C.); huangwendy227@gmail.com (Y.-W.H.); vincent.lee67@gmail.com (W.-C.L.); 2Industrial Technology Research Institute, Hsinchu 31040, Taiwan; 3Research Center for Applied Sciences, Academia Sinica, Taipei 114699, Taiwan; loveriver031@gmail.com (Y.-Y.L.); yjcheng@sinica.edu.tw (Y.-J.C.); 4Taiwan Semiconductor Research Institute, Hsinchu 30078, Taiwan; jerrytzou.ep00g@gmail.com; 5International College of Semiconductor Technology, National Yang Ming Chiao Tung University, Hsinchu 30010, Taiwan; edc@mail.nctu.edu.tw; 6Semiconductor Research Center, Hon Hai Research Institute, Taipei 114699, Taiwan

**Keywords:** gallium nitride, high-electron mobility transistor, heterogeneous integration, SOI, QST

## Abstract

GaN HEMT has attracted a lot of attention in recent years owing to its wide applications from the high-frequency power amplifier to the high voltage devices used in power electronic systems. Development of GaN HEMT on Si-based substrate is currently the main focus of the industry to reduce the cost as well as to integrate GaN with Si-based components. However, the direct growth of GaN on Si has the challenge of high defect density that compromises the performance, reliability, and yield. Defects are typically nucleated at the GaN/Si heterointerface due to both lattice and thermal mismatches between GaN and Si. In this article, we will review the current status of GaN on Si in terms of epitaxy and device performances in high frequency and high-power applications. Recently, different substrate structures including silicon-on-insulator (SOI) and engineered poly-AlN (QST^®^) are introduced to enhance the epitaxy quality by reducing the mismatches. We will discuss the development and potential benefit of these novel substrates. Moreover, SOI may provide a path to enable the integration of GaN with Si CMOS. Finally, the recent development of 3D hetero-integration technology to combine GaN technology and CMOS is also illustrated.

## 1. Introduction 

### 1.1. History and Applications of GaN HEMT

In the past decades, the wide bandgap GaN semiconductor materials and its alloys (AlGaN and InGaN) are emerging as one of the most promising materials for a variety of applications. Due to its robust thermal stability and electronic properties such as radiative hardness, it is an ideal candidate for working in a harsh and aggressive environment [1,2]. The development of AlGaN/GaN high electron mobility transistors (HEMT) is mainly due to both military and commercial interests in high-temperature, high-frequency, and high-power device applications [2,3,4]. AlGaN/GaN HEMT has been extensively developed for radio frequency (RF) high power amplifiers with high output power and high efficiency [3,5,6,7]. GaN’s high breakdown electric field (~3.3 MV/cm) and high mobility (>900 cm^2^ Vs) [8] makes GaN-based devices attractive to work in very high powers and microwave frequencies since it can handle high current (~ 10 A) and high voltage (~ 100 V) with high transit speed [9,10]. The first GaN RF device product was presented in 2005 [11] but the insertion of the technology was limited in the military and some high-end RF infra-structure applications due to high cost. Later on, the first commercial product of GaN HEMT for power electronic application was presented in 2009 [12]. Owing to its superior performance and manufacturability on Si, GaN HEMT enters the stage of rapid growth. More recently, GaN HEMT has been rapidly developed for high-performance RF power amplifiers (PAs), such as mmW PA for 5G and beyond applications, owing to the requirement of wide-bandwidth and data rate for future mobile communication systems [13]. The roadmap in GaN HEMTs applications is illustrated in Figure 1.

### 1.2. The Influence of Different Substrates on GaN HEMT Device

In the earlier stage of its development, the progress of GaN/AlGaN HEMT was focused on improving the quality of epitaxial material for RF applications, the major efforts include the selection of the best substrate as well as developing unique processes [14]. GaN HEMTs can be grown on different substrates, including sapphire, silicon (Si) silicon carbide (SiC), or diamond due to the lack of GaN bulk substrate [15,16]. The basic structure of GaN HEMT is shown in Figure 2. For RF GaN HEMT, it was expected to provide a very high output RF power for a single die up to several hundred watts over a wide frequency range [17,18]. The high-power density requires efficient power dissipation on transistors as well as on substrate [19], thus, the performance and reliability of the high-power RF transistor can be seriously affected by the thermal conductivity of the substrate. The thermal properties of the different substrates are shown in Table 1. By using a substrate with high thermal conductivity, like SiC, the high heat generated can be effectively dissipated. AlGaN/GaN HEMTs on SiC substrates with very high RF output power densities of about 40 W/mm have been reported [20]. However, the full advantages of GaN on the high thermal conductivity SiC cannot be obtained as the thermal performance is degraded due to the epitaxial layer defects at the interface between GaN and SiC [21], which we will discuss further in a later paragraph. Cree has been developing GaN on SiC for commercial sale since 2006 [22]. The gradual scale-up of high purity semi-insulating 4H-SiC substrates from 2-inch to 4-inch substrates has greatly enhanced the economic viability of wide band-gap microwave devices.

In the past, SiC is mainly used as a substrate for RF GaN HEMT and demonstrated satisfactory performance. However, Si substrate is desired due to its low cost, good thermal conductivity, and availability of a large area. The growth of GaN on Si is more difficult than the growth of GaN on SiC since the growth of GaN on Si tends to result in higher dislocation density or microcracks due to higher thermal expansion coefficient and lattice mismatches [23]. With the progress of material engineering, the low cost and high performance. GaN on Si device has been achieved on a high-quality GaN layer grown on a large area Si substrate (8-inch wafer) [24]. One of the main issues of fabricating RF GaN HEMT on Si is the increased RF losses due to the parasitic conduction channel introduced at the III-nitride–Si interface and lower resistivity of Si substrates [25,26,27,28]. High resistive (HR) Si substrates are often adopted to solve the issue. However, the growth of high strain GaN on HR Si substrate shows a worse bowing problem. Moreover, the cost of HR Si is higher than that of low resistive (LR) Si substrate. These factors limit the production of the device on a larger silicon substrate. Substrate removing technique is reported to allow the fabrication of GaN HEMT on LR Si substrate with minimized substrate parasitic to improve RF/MW characteristic [29]. Though high-performance RF GaN HEMT on Si is still in development, GaN HEMT on Si has become the mainstream for power electronics applications. Currently, the power GaN HEMT with breakdown voltage higher than 1200 V has been demonstrated. In the later paragraphs, we will discuss and review the status of GaN on Si devices for both RF and power applications in details.

Among all the potential substrate materials, single-crystal diamond shows the highest thermal conductivity [30]. A high cut-off frequency (85 GHz) of GaN HEMT on the diamond was fabricated using wafer bonding between GaN HEMT structure grown on Si wafer and polycrystalline diamond wafer [31,32]. AlGaN/GaN HEMT structure grown by molecular beam epitaxy (MBE) on diamond (111) was reported with RF small-signal characteristic [31,32]. A single-crystal AlN and AlGaN/GaN HEMT were grown on a diamond substrate by using (111) surface orientation with metal-organic vapor phase epitaxy (MOVPE) [15,33,34]. AlGaN/GaN HEMTs grown on diamond (111) substrate for RF power operation with an output density of 2.13 W/mm were achieved [31]. Although the device performance for GaN on diamond looks promising, the lack of a large diamond substrate limits the development of the technology. Commercially available sapphire is also one of the most prominent substrates for the epitaxy of GaN due to the limited native GaN availability [35]. However, the use of sapphire is limited as the stability of the GaN on sapphire is less than that of GaN on Si due to weaker scattering phenomena which is related to its thermal conductivity at high drain bias [36].

Along with the thermal conductivity of the substrate, Thermal Boundary Resistance (TBR) is an important parameter to affect the overall rise in temperature of a device [37]. The thermal effects of the device are boosted by TBR which is resulted from the interface for the growth of dissimilar materials. TBR is a measure of the resistance imposed by the interface to heat flow due to the different dynamics of the phonons and the poor quality of the crystal near the boundary. It is defined as the maximum temperature increase by the maximum power dissipation of the device. The TBR in commercial GaN HEMTs on SiC can reach levels greater than 6 × 10^−4^ cm^2^K/W which can increase the maximum temperature of the device by up to 40–50% [38,39]. The increase in TBR causes an offset to the advantages offered by the substrate with high thermal conductivity due to the discontinuity of the temperature gradient in TBR layers located at the GaN substrate interface [40]. Thermal conductivity for GaN and different substrate materials at 300 K and TBR values of GaN/ substrate interface simulated in [41] are listed in Table 1. The lower TBR values of GaN on Si substrate than that of SiC substrate is mainly due to the different thermal expansion coefficient [42], the roughness of the substrate materials as well as the defect related to the growth techniques. The high value of TBR at the interface of high thermal conductivity substrate and GaN could be reduced and has additional advantage if it has a better thermal coupling and fewer interface defects while low thermal conductivity like sapphire has a less significant influence of TBR on transporting heat [43].

### 1.3. Evolution of GaN HEMT

#### 1.3.1. Different Gate Structure Designs

The HEMT structure was based on T. Minura et al. (1975) [45] and M.A. khan et al. (1994) [5]. At the interface of AlGaN and GaN, there exists 2-dimensional electron gas (2DEG) with high electron mobility owing to the difference in spontaneous polarization and piezoelectric polarization [46]. Hence the device operates as a normally-on device naturally. To deplete the 2DEG channel, a gate electrode on top of the AlGaN layer has a negative gate voltage concerning drain and source electrode applied. This type of device is known as depletion-mode (D-mode) HEMT. Moreover, there are two types of D-mode HEMT, namely with a Schottky gate electrode or with an insulating gate [47]. The first d-mode HEMT introduced had a Schottky gate electrode in which the metal gate electrode is directly deposited on top of AlGaN. Ni–Au or Pt metals were used to form the Schottky barrier [23,48,49]. In insulate gated d-mode HEMT, an insulating layer is placed in between the gate electrode and AlGaN similar to that of MOSFET to block the gate current [50]. Schottky gate and insulated gate d-mode HEMTs are shown in Figure 3.

D-mode HEMT is not preferred in system applications as it requires a negative bias to be applied. On the other hand, there is also a concern in fail-safe operation. Therefore, Enhancement-mode (E-mode, normally-off) device is favored and has become one of current focuses of technology development. To create e-mode devices, there are five popular structures: recessed gate, implanted gate, pGaN gate, direct-drive hybrid, and cascode hybrid [47]. For GaN HEMT in RF applications, the most common fabrication technique used for modifying the threshold voltage is “gate-recess”. This process reduces the barrier thickness under the gate metal. The basic structure of gate recess HEMT is shown in Figure 4a.

There are also approaches to obtain E-mode devices by heterostructure or gate stack designs. Ohmaki et al. proposed a double-barrier-layer AlGaN/GaN HFET in 2006. The use of double-barrier-layer sustains the device at a BV of 435 V, Ron,sp of 1.9 mΩ cm^2^, with a threshold voltage of −0.1 V [51]. Inter University Microelectronics Centre (IMEC) presented another E-mode AlN/GaN/AlGaN HFET structure in 2010 with a double heterostructure-barrier layer. The double structure comprises of a high concentration 2DEG at the surface of the hetero structure and an ultrathin AlN barrier layer grown on silicon substrate; and the structure maintains a BV of 580 V, with an Ron,sp of 1.25 mΩ cm^2^, and a threshold voltage of about 0 V [52]. Mizutani et al. in 2007, proposed an E-mode AlGaN/GaN HFET with a thin InGaN cap layer. The proposed structure shifts the threshold voltage towards the positive direction by rising the conduction band of the AlGaN/GaN via the use of a polarization-induced field in the InGaN cap layer [53]. Ostermaier et al. in 2009, proposed an ultrathin InAlN/AlN barrier HEMT that have high performance under normally off operation. They selectively etched the n++ GaN cap layer structure, which in-turn controls the width of the ultrathin barrier layer [54]. The proposed structure operates with a threshold voltage of 0.7 V, a maximum transconductance of 400 mS/mm and a maximum output current of 800 mA/mm. Hughes Research Laboratories, in collaboration with the Next Generation of Nitrides Electronics Project from the US Defense Advanced Research Projects Agency (DARPA), proposed an E-mode HFET with a double-barrier layer via selective-area molecular beam epitaxy (MBE) [55]. For the integrated E-mode and D-mode AlN/GaN/AlGaN double-heterojunction field-effect transistors (DHFETs) on a single SiC substrate, an E-mode channel was achieved without additional procedures or compromise in the electrical characteristics. The idea of developing a monolithic integrated E-mode and D-mode device can be the groundwork for direct-coupled field-effect transistor (FET) logic circuits. Guowang Li et al. proposed an E-mode AlGaN/AlN/GaN HFET having 70% Al composition comprising of a thick AlGaN layer (17 nm), an AlN layer (0.6 nm) and an Al_2_O_3_ (4 nm) layer. They used Ni/Au in the Schottky gate metal instead of Al/Au thereby enhancing the threshold voltage from −1.0 V to −0.13 V [56]. Chiu et al. proposed an E-mode HFET comprising of a composite dielectric layer via N_2_O plasma oxidation technology. The composite dielectric layer (Al_2_O_3_/Ga_2_O_3_) after the oxidation in the N_2_O for the AlGaN barrier layer improves the threshold voltage from −3.6 V to 0.17 V [57]. Based on this structure, Chiu et al. presented an E-mode HFET with a high-*k* composite dielectric layer in 2012. The AlGaN barrier layer was oxidized by nitric oxide (NO) gas prior to the Schottky metal gate deposition. After the oxidized AlGaN barrier layer stack (Al_2_O_3_/Ga_2_O_3_) is formed, a dielectric film of gadolinium oxide (Gd_2_O_3_) was deposited to complete the structure thereby, enhancing the threshold voltage from −3.15 V to 0.6 V [58]. Massachusetts Institute of Technology in collaboration with Harvard university presented one E-mode HFET with a scandium oxide (Sc_2_O_3_) high-*k* dielectric. In the study, Wang et al. introduced a Sc_2_O_3_ high k-dielectric layer in the HFET to reduce the invert leakage current thereby having the switch-current ratio (*I*_ON_/*I*_OFF_) reach 10s [59].

By reducing AlGaN barrier thickness, it can result in a reduction of polarization induced 2DEG and the threshold voltage is shifted positively with the help of the work-function of the metal gate. A positive threshold voltage can be achieved with a deep enough gate recess etching thus forming an E-mode HEMT [60]. A chloride-based dry inductively coupled plasma reactive ion etching (ICP-RIE) for gate recess etching has been employed by several groups [61,62,63,64,65,66] which can effectively change the threshold voltage to the positive direction of AlGaN/GaN HEMT. Damage in the subsurface was also reported from ICP dry etching due to low etch selectivity between materials which increases the gate-leakage current [67]. The damages were found to be repaired after post-etching rapid thermal annealing (RTA) at 700 °C [65,66]. For the RF HEMT structure, recessed gate enhances the device performance by providing a better gate control capability of the channel carriers. On the other hand, due to the repaired damage with proper recessed gate process, flicker noise characteristics were reduced, which is beneficial for RF circuit applications such as voltage-controlled oscillators (VCOs) and mixers [68,69]. On the other hand, p-GaN gate technology is more popular for the fabricator of e-mode power GaN HEMT device. The schematic structure of p-GaN gate HEMT is shown in Figure 4b. The conduction band of the AlGaN is risen above the Fermi level when compared with a standard Schottky gate normally-on HEMT due to the presence of p-GaN cap and results in the depletion of the 2DEG channel. Uemoto et al. first proposed a p-AlGaN gate normally-off HEMT [70]. The thickness of the AlGaN barrier as well as Al-concentration needs to be defined appropriately to achieve an efficient depletion region in the channel [71,72,73,74]. A high doping concentration of Mg, a p-type dopant, (>10^18^ cm^−3^) is necessary for the efficient depletion at the interface of the metal gate/p-gate [75]. However, the hole concentration can be reduced at a high temperature above 500 °C due to the formation of Mg-H complexes [76]. Hence, proper care is necessary during device processing like annealing of Ohmic contacts so that Mg concentration does not reduce. Another important feature is the choice of the metal gate as the threshold voltage of the device depends on metal/p-GaN Schottky barrier height. In this aspect, much researches have been done on metal gate work-function influence on p-GaN electrical behavior [77,78,79,80]. Schottky metal gates have shown improvement in lowering the leakage and increasing the threshold voltage than Ohmic gate [77,78]. Presently, the TiN gate is one of the good solutions due to its thermal and chemical stability along with the processing compatibility [81,82,83,84]. A p-GaN gate HEMT with a “self-aligned” process using Mo-based gate was demonstrated which employed “gate first” process [85]. Mo gate sustained the high-temperature annealing process of source-drain ohmic contact without the barrier degradation. Although p-GaN gate HEMT has reached commercialization, their reliability issues [80] need intensive researches.

#### 1.3.2. N Polar vs. Ga Polar

Typically group III-Nitride devices are fabricated using the Ga-polar (0001) orientation. However, the inverted N-polar polarity possesses a numerous advantage over Ga-polar counterparts. The absence of inversion symmetry in wurtzite group III-Nitride results in opposite polarization of N-polar crystal and Ga-polar crystal. Hence, the polarization induced electric fields of Ga-polar heterostructures is opposite to that of N-polar counterparts which results in formation of 2DEG of N-polar heterostructures above the wide-bandgap barrier layer instead of below [86]. The advantages offered by N-polar GaN HEMTs over Ga-polar HEMTs are as follows: (i) N-polar heterostructures has a strong back-barrier due to its inherent wide-bandgap Al(Ga)N back-barrier for electron confinement which reduces the effects due to short-channel [87], (ii) N-polar HEMTs has low-resistivity Ohmic contact as the channel layer with lower surface barrier to electrons and a narrower bandgap can contact 2DEG of N-polar HEMTs rather than contacting through wide-bandgap Al(Ga)N barrier [88,89] which results in the possibility of lowering the contact resistance by using selective regrowth of ohmic area in N-polar structure, (iii) N-polar heterostructure has improved scalability. The effective gate-channel distance is reduced by the quantum displacement of the 2DEG in N-polar HEMT which is opposite to that of Ga-polar HEMTs owing to the reduction in effective gate-channel capacitance due to quantum capacitance [90]. Such enhancement results in higher N-polar transconductance than that of Ga-polar with same gate-channel thickness. The aspect ratio as well as the charge density under the gate of N-polar HEMTs can be controlled independently where the enhancement of charge depletion due to the scaling of channel thickness are compensated by increasing the thickness and the polarization of charge-inducing back barrier. Whereas, gate aspect ratio in Ga-polar HEMT depends on the barrier thickness and there is a trade-off between the charge density under the gate and the barrier thickness [91]. N-polar heterostructure does not required intentional n-type doping for the formation of 2DEG [89,92,93] due to the presence of high-density unintentional bulk and surface donors [94]. A N-polar GaN HEMTs epitaxy grown by MOCVD on 4° off-cut 2-inch diameter sapphire substrate at UCSB for X-band power performance transceiver systems was reported with 2 tone results at 10 GHz [95]. The reported HEMTs structure is similar with that of structure reported by X. Zheng et al. [96]. The reported HEMT device shows very little 3rd order distortion in intermodulation and 65% of single tone high power added efficiency (PAE) with 3 W/mm power density which can be scaled favorably with 15 V drain voltage. A deep recess W-band power N-polar GaN HEMTs utilizing a new atomic layer deposition (ALD) ruthenium (Ru) gate metallization process with PAE of 33.8% and 6.2 W/mm high power density was demonstrated [97]. The demonstrated HEMTs has an outstanding control over the DC-RF dispersion due to presence of deep recess structure in conjunction with SiN thin passivation layer and has a high gain as the narrow 48-nm gate trench was filled by ALD Ru metal.

### 1.4. CMOS Compatible Process for GaN HEMT

In wireless communication, to respond to the growing demand for high data rates, there is a push to a higher operating frequency, switching to millimeter-wave from the congested sub 6 GHz band. Beyond operational speed of power amplifiers, output power (P_out_) and power added efficiency (PAE) in RF Front End Modules are critical for next-generation portable devices and small cells. As opposed to CMOS, the high-power handling capabilities of GaN are advantageous for mm-wave operations [98] as the more energy-efficient system can be achieved owing to its capability in the high output power with high efficiency at high-frequency. However, the integration of GaN HEMTs remains one of the main concerns. The current limitation of GaN RF technology is due to the use of expensive older generation Au-based processing as well as non-Si substrates [99]. Hence to make GaN devices for RF and MM-wave applications, migrating to 200 nm Si platform and using standardized CMOS fabrication tools for manufacturing the device become a crucial step to improve yield and reduce the cost. Similarly, the CMOS compatible technology can also benefit the development of GaN devices for power electronics. A CMOS-compatible 110 V/650 V e-mode GaN HEMT with excellent power converter switching performance with high robustness was fabricated on 6-inch GaN on a Si wafer [100]. Figure 5 shows an example fabrication process flow of GaN devices. GaN devices with low RF loss, low buffer dispersion as well as good leakage blocking capability have been demonstrated by integrating the device on the Si platform based on the Au-free, Si CMOS compatible process [98,101,102,103]. To enhance the functionality as well as the performance of the RF modules, various approaches to integrate CMOS and GaN devices have been developed [104,105].

### 1.5. GaN-Based CMOS Technology

A lot of demonstration has been done on power integrated circuit based on GaN [106,107,108] which depends on the integration of E-mode and D-mode n- type HEMTs. However, these E/D mode circuits have reduced output voltage swing and also suffers from static power dissipation. Therefore, a CMOS-like circuit technology is required to increase the efficiency of ICs based on GaN as this circuit technology has negligible static power consumption, has higher noise immunity and less circuit complexity [109]. However, the challenges of the monolithic integration of the p-type GaN FETs with n-type GaN FETs along with lack of high-performance of p-type GaN FETs are the major obstacle towards achieving high efficiency GaN-based ICs. A various epitaxial structures of p-type GaN FETs have been demonstrated [110,111,112,113,114,115,116,117]. A GaN complementary inverter circuit comprising of both E-mode n-type GaN FET and p-type GaN FET monolithically integrated on Si Substrate without regrowth technology was also demonstrated [118]. The probe station with thermal chuck were used to characterize the fabricated inverter under high operation temperature. The fabricated circuit shows an outstanding transfer characteristics up to 300 °C with maximum recorded voltage gain of about 27 V/V at 0.59 V input voltage with 5 V as V_DD_ supply. A very high density of 2D hole gas (2DHG) induced by the polarization at the interface of GaN/AlN was discovered [119] which led to the development of p-channel heterostructure field effect transistors (HFETs) that reach the linear current density of 100 mA/mm [112]. A p-channel MISFET with a recessed-gate was grown by metalorganic chemical vapor deposition (MOCVD) using p-GaN/AlGaN/GaN hetrostructure on Si substrate [113,118]. The fabricated structure contains both 2-dimensional electron gas (2DEG) and 2-dimensional hole gas (2DHG) without regrowth technology which is suitable for implementing GaN- based complementary circuit. The fabricated long channel p-type device when compared with p-FET GaN/AlGaN on sapphire substrate exhibits state of the art on-off ratio performance. A p-channel 2DHG GaN/AlN transistors which can break the barrier of GHz speed was demonstrated [120]. The fabricated transistor exhibits an on-current density of 428 mA/mm and a cut-off frequency of 20 GHz. A wide-bandgap CMOS platform formed using these fabricated p-channel HFETs along with excellent performance n-channel HFETs [121] was expected to achieve a new domain in the RF and power electronics applications [122].

In the later paragraphs, we will review in more detail the progress and performance of GaN on Si devices. Moreover, the progress in hetero-integration will also be described.

## 2. GaN HEMT on Si

### 2.1. GaN Epitaxial Growth on Silicon

Silicon substrate is a general and commercial materials in semiconductor technology. Actually, a good epitaxy must be fine-matched in lattice and thermal expansion coefficient between GaN and heterogenous substrate, like Si, Sapphire, SiC…etc. The related summary in physical parameters is listed in Table 2 [123]. 

Due to the mismatch of lattice constant and thermal expansion coefficient between Si and GaN, it is more difficult to achieve high quality GaN structure growth on Si [124,125,126]. In general, a lot of defects exist, and the gallium nitride layer may crack when cooling process. In order to solve these problems, gallium nitride epitaxial layers with high uniformity, high quality and no cracks can be grown on silicon substrate by growing buffer layer as shown in Figure 6 [12,127]. The buffer layer can be designed in the following two ways: (1) AlN/GaN superlattice growth through the step gradient AlGaN layer [128], and (2) AlGaN/GaN superlattice [129]. The bending of the wafer will increase with the increase of the thickness. A multilayer-buffer structure reduces the tensile stress caused by the huge difference in thermal expansion coefficient between GaN and Si. It’s useful for inducing compressive strain in the growth process, as to counteract tensile strain introduced in the cooling process, prevent cracking and produce a flat wafer.

Furthermore, the breakdown voltage in GaN HEMT is affected by the quality and resistivity of GaN-based templates. In order to operate an efficient GaN channel, it also needs higher resistivity buffer layer to prevent the DC leakage current and AC coupling. The characteristics of GaN high-frequency power amplifiers will change with the resistance of the underlying buffer layer, which is mainly due to the signal coupling effect. However, several groups have used different methods to improve the buffer layer resistivity, which employed p-type dopant (Mg) to enhance GaN buffer structure (n-type intrinsically). Another method is using Carbon dopant which plays a more attractive role in buffer layer. Compared to Mg case, the storage effect of Carbon-doped method isn’t strong. According to doped buffer layer, the carrier concentrations and electrical properties (breakdown voltage) are adjusted by varied epitaxial conditions [130].

On the other hand, the top barrier layer includes AlGaN or InAlN, which results in 2DEG of HEMT through polarization charges in nitride-based materials are critical. While the thin AlGaN layer is grown on GaN channel layer, the Al content and thickness would be limited due to the lattice mismatch of GaN. The interface charge could be adjusted by varied barrier thickness and Al compositions. Compared to AlGaN, InAlN could reduce epitaxial defects as a result of thicker critical thickness. It shows that a good lattice-matched InAlN (18% In) exhibits a stronger spontaneous polarization to generate a higher channel charge density [131]. 

### 2.2. Power GaN Performance Si Substrate

As described in Section 1, the traditional power HEMT structure uses a Schottky metal to modulate the 2DEG in the channel. Generally, the metal stack Ni/Au is used for the HEMT. However, in order to efficiently control the gate leakage current, high-k gate dielectric layer was employed to form metal–insulator–semiconductor (MIS) gate, [132] for commercial power GaN HEMT on Si. The passivation layer provides additional protection and reduces the current loss in the surface state of devices. The breakdown voltage of GaN power components on Si in the low- and medium-power fields are predicted above 900 V, and the GaN shows very high potential in power applications due to the benefits of a low switching loss and lower cost [133].

Furthermore, the power electronics application represents one major market in the developments for GaN power devices like p-GaN HEMTs for enhancement-mode (E-mode) operation. The breakdown voltages for p-GaN HEMTs have exceeded 1000 V (Ron,sp of 2 mΩ cm^2^). In power switching applications, a normally-off (enhancement-mode) GaN HEMT is desirable due to the safe-operation formation and efficient gate control to switch on/off. A variety of e-mode GaN HEMTs are fabricated by using p-GaN gate [74], gate recess [134], or plasma treatment techniques. The p-GaN gate HEMT showed a good performance, reliability, and commercialization. Figure 7 shows the band structures of normally-on AlGaN/GaN HEMTs and normally-off p-GaN/AlGaN/GaN HEMTs. The 2DEG channel is depleted at a zero-bias condition as the conduction band energy of AlGaN is lifted due to the p-GaN region. The electrical characteristics of the p-GaN gate HEMT shows V_TH_, the V_GS_ limitation, and the gate leakage current (I_GSS_) depend on the structure of the gate stack by using normally on and off system [135]. In order to deplete the 2DEG channel at V_G_ = 0, the general AlGaN thickness is 10~15 nm, and the thickness of the p-GaN gate is 50~100 nm. A doping Mg concentration of the p-GaN (or p-AlGaN) gate is around 10^18^~10^19^ cm^−3^.

### 2.3. RF GaN Performance Si Substrate 

GaN technology that can offer high output power and efficiency at high frequencies is regarded as the most critical technology to reduce the complexity in designing upcoming mm-wave band communication system for 5G or beyond applications. In particular, GaN on Si technology that can greatly reduce production costs has attracted more attention. However, due to the challenges in obtaining higher-quality epitaxy on Si, RF GaN HEMT was fabricated primarily on SiC for high-frequency applications until 2014 when MACOM announced the mass production of 4” GaN on Si technology. Due to the continuous improvement of epitaxial technology, almost all major semiconductor foundry starts to invest heavily on the development of GaN on Si technologies.

In order to develop RF GaN HEMT with superior high-frequency characteristics, we can refer to the equivalent circuit [136] shown in Figure 8, the small signal model of a GaN HEMT includes gate parasitic capacitance and resistances. According to Eq (1)(2) [137], we must decrease capacitance [138], ohmic contact resistance [139], gate resistance [140] and increase transconductance [141] in order to maximum the f_T_ and fmax.
(1)$${\;F}_{T}=\frac{g_{m}}{{2\pi (C}_{\mathrm{GS}}{+C}_{\mathrm{GD}})[1+\left(R_{S}{+R}_{D}\right)G_{\mathrm{DS}}{+g}_{m}{\times C}_{\mathrm{GD}}{(R}_{S}{+R}_{D})]}$$
(2)$$F_{\mathrm{MAX}}=\frac{F_{T}}{2\sqrt{\left(R_{i}{+R}_{S}{+R}_{G}\right)\times G_{\mathrm{DS}}{+2\pi F}_{T}R_{G}C_{\mathrm{GD}}}}$$

To operate at high frequency, the gate length must be minimum to reduce gate capacitance, as shown in Figure 9. But, the parasitic resistance will be increased and degrade the high frequency performance [142], the T-shaped gate becomes a key element to reduce the gate parasitic resistance. The height and width of the T-shaped gate should be optimized for both capacitance and resistance values. Keisuke Shinohara et al. [142] demonstrated the most suitable T-gate shape through simulation based on gate capacitance as show in Figure 10 and Figure 11. Benchmark of cut-off frequency versus L_G_ [143,144,145,146,147,148] illustrates the importance of gate length shrinking to increase f_T_. Both NTU [149] and Intel [143] demonstrated 40 nm gate length GaN HEMT on Si with f_T_/fmax higher than 300 GHz, though the current record f_T_/fmax values of 450 GHz was achieved by HRL [144] with the 20 nm Gate GaN on SiC technology.

There are many device technologies have been developed based on GaN HEMT on SiC substrates. Most of those technologies can be applied on GaN HEMT on Si as well. We will describe some of the typical examples below. Hiroyuki Ichikawa et al. [150] designed 150 nm gate length InAlN/GaN and AlGaN/GaN HEMT, the AlInN/GaN HEMT showed high G_M_ and exhibited a f_T_/fmax of 70/150 GHz. Michael L. Schuette et al. [141] demonstrated a peak f_T_/fmax of 348/340 GHz for 27 nm gate length on InAlN/GaN HEMT with gate recess. Ezgi Dogmus et al. [151] used ultra-thin AlN (4 nm) barrier to replace AlGaN and in-situ SiN demonstrated a f_T_/fmax of 55/235 GHz. Jeong-Sun Moon et al. [152] designed 50 nm gate length AlGaN/GaN HEMT with graded AlGaN barrier and n++regrowth, and the HEMT exhibited a ft/fmax of 156/308 GHz. Lei Li et al. [153] demonstrated f_T_/fmax of 250/204 GHz using n++regrowth for InAlN/GaN HEMT on Si substrate. Figure 12 benchmark f_T_ versus f_MAX_ [141,150,151,152,153,154,155]. 

GaN HEMT on Si also have high load pull result comparable to SiC substrate [155], nevertheless, GaN HEMT on Si substrate shows high potential. D.C. Dumka et al. [156] demonstrated 13.1 dB linear gain, maximum P_OUT_ = 34.5 dBm, output power density 7 W/mm and PAE 65.6% at 10 GHz in X band. Diego Marti et al. [157] showed 6 dB linear gain, output power density 1.35 W/mm, and PAE 12% at 94 GHz in W band.

For used in 5G mm-Wave communications, higher data rates require more complex communication systems such as Multi-input Multi-output (MIMO). MIMO employs complex frequency and phase division. So high linearity devices are required to avoid interaction each complex frequency bands. J. Vidkjær et al. [158] summarized some solution for linearity include geometrical, layout and epitaxial design. Weichuan Xing et al. [159] designed 150*150 nm nanostrip gate hole structure by BCl_3_/Cl_2_ and Al_2_O_3_ insulator which have good linearity. Jeong-sun Moon et al. [160] used AlGaN/GaN graded channel which have good PAE and linearity. Bin Hou et al. [161] used barrier layer of sandwich structure and AlGaN back barrier which show good power performance and linearity. Kai Zhang et al. [162] used Fin-FET HEMT that have good linearity compared to the planar HEMT.

## 3. GaN HEMT on Silicon-on-Insulator (SOI) Substrates 

### 3.1. GaN Epitaxial Growth on Silicon-on-Insulator (SOI) Substrates

In epitaxial issues, the bowing effect always exists on the hetero-interface due to the lattice mismatch and the thermal expansion differences. T. Egawa from Nagoya Institute of Technology has reported a relative function of the wafer bowing and epitaxial thickness of AlGaN/GaN HEMT on Si [163], as shown in Figure 13. A thicker GaN/AlN superlattice structure exhibits higher bowing value in these experiments. It is a wafer bowing reference in HEMT epitaxial developments on Si. However, based on the outstanding electrical isotropic and mechanical features of the SOI substrate, it is expected to be a significant contender as a technological platform for mass production of GaN HEMT in the near future. However, the SOI substrate still suffer from a bowing effect, which may result in broken wafers or difficulties in subsequent fabrication processing steps, as well as a lower temperature tolerance during wafer process. Recently, from our study, we demonstrate the growth of AlGaN/GaN heterostructure on a 150-mm SOI substrate with different boron doping concentration in handle wafer, as shown in Figure 14. By heavily doping Boron in silicon handle wafer of SOI substrate, we can effectively reduce the bowing effect, increase the thickness of the epitaxial layer, and further improve the device performance. Heavily doped handle wafer causes a reduction in wafer bowing by >97%, as shown in Table 3. Moreover, it can be seen from Figure 15 that the issue of edge cracks for the heavily doped SOI substrate (sample B) are great improved, and there is no peeling phenomenon, which means that the heavily doped SOI substrate have better ability to resist the stress generated during GaN epitaxy. Figure 16 is the comparison of the half-width values of the GaN epitaxial layer (102) measured by X-ray diffraction analyzer. It can be observed that the BOW value is related to the epitaxial quality. That is, the more severe the bowing effect, the worse the epitaxial quality, which has also been demonstrated in GaN on SiC and bulk GaN substrate [164,165].

### 3.2. Power GaN HEMT on SOI

In the past several years, enhancement mode (E-mode) AlGaN/GaN HEMTs have been demonstrated to be the potential devices for next generation high efficiency power switches and converters application [2,166]. Currently, SiC [19] and Si [167] are the most popular substrates for GaN HEMT. However, GaN-on-SOI has been considered as a highly potential option which may provide better performance in high frequency and high-power system, owing to its capability in defect reduction of epitaxial layer, as described in Section 3.1.

According to our studies shown in Section 3.1, the BOW value of GaN on SOI substrate is close related to threading dislocation density (TDD) and epitaxial quality. The HEMT devices on SOI substrate with lower BOW value exhibit 1 order smaller off-state leakage and 8.4% smaller specific on resistance, also 68.8% improvement is observed in 3-terminal off-state breakdown voltage (BV_GD_), shown as Figure 17. Moreover, the dynamic R_ON_ degradation can be reduced. This implies that by heavily doping in handle wafer not only reduce the bowing effect, but also improve the quality of substrate and the performance of high-power device.

GaN-on-SOI substrate exhibits a capability to improve the power device performance and also have been proven by many research teams [168,169,170,171,172] including smaller reverse recovery leakage [169], higher breakdown voltage [168,169] and smaller vertical leakage [172]. Kevin J. Chen et al. [169] reported the SOI substrate with a reduced stress in the GaN epilayers (shown as Figure 18a) and an excellent E-mode HEMTs on SOI produced by fluorine plasma implantation method with a high ON/OFF current ratio (10^8^–10^9^), large breakdown voltage (1471 V with floating substrate), and also a smaller vertical leakage at reverse bias, as shown in Figure 18b.

Moreover, most GaN power switching systems are currently produced using a multi-chip approach, resulting in significant complexity and expense [173,174,175]. Monolithic integration of GaN-based power devices has steadily gained interest for GaN high-power systems. The benefits of monolithic integration of GaN power systems on a single chip include minimizing parasitic inductance, reducing die size, and enhancing design flexibility [107,176]. To prevent mutual influence between the devices in monolithic GaN power integrated circuit, it suggests that the low and high side HEMT transistors must be fully isolated for a half bridge, as shown in Figure 19 [172,177]. However, it is challenging to accomplish on GaN-on-Si substrates because those HEMTs share a common conductive Si substrate. Figure 20a shows the transfer characteristics of a GaN monolithic half bridge with a common Si substrate biased from −200 to 200 V at 150 °C. Significant variations in threshold voltage (Vth) and drive current are seen when the Si substrate is biased negatively. Nevertheless, using GaN-on-SOI and a trench isolation method, this issue could be overcome [172,177,178]. The transfer characteristics of GaN-on-SOI HEMT, as shown in Figure 20b, illustrate the benefits of device isolation. When the substrate of a neighboring device is biased between −200 V and 200 V, transfer characteristics vary relatively little, which is in sharp contrast to characteristics on a silicon substrate.

### 3.3. RF GaN HEMT on SOI

In addition to the power devices, AlGaN/GaN HEMTs have also been demonstrated to be the potential devices for RF applications [179,180,181,182]. Moreover, GaN-based MMIC (Microwave Monolithic Integrated Circuit) has steadily gained interest as compared to a system in package or a multi-chip module, since monolithic integration of GaN RF systems allows for smaller, cheaper, and less complicated circuitry [181,182].

SOI substrate is outstanding for its better vertical isolation performance and a lower substrate loss [183]. Besides, when compared to GaN on Si substrate devices, GaN on SOI substrate devices demonstrated better DC, breakdown voltage, and RF properties [170,184,185]. It was demonstrated that GaN-on-SOI substrates perform better in terms of tensile stress relaxation and surface flatness than Si substrates. It can result in a reduction of defect density, which is further supported by pulse and low-frequency noise measurements. The SOI substrate capacitances extracted from the small signal model are lower than the HR-Si substrate, as illustrated in Figure 21, owing to the series connection of device layer, buried oxide layer, and handle wafer of SOI substrate. As a result of the small substrate capacitances of SOI substrate shunt to the C_DS_, the effective C_DS_ items that dominated the feedback capacitance were decreased. Furthermore, the utilization of SOI substrate can increase the device’s bandwidth and linearity was proven at the same time, as shown in Figure 21b [185].

## 4. GaN HEMTs on QST Substrates

Recently, a new engineered substrate consists of polycrystalline core and single crystalline surface layer, which exhibits a good thermal expansion and crystalline match to GaN, are presented. It provides a good matching in coefficient of thermal expansion (CTE) characteristics with Gallium Nitride (GaN). The template enables growth of thick and high quality GaN semiconductor layers on 8- and even 12-inch wafers and support a lower cost for GaN devices in power supplies and RF transmitters commercial markets. According to Qromis Inc., Qromis Substrate Technology (QST) promises a thicker GaN epitaxy to expands the GaN HEMT’s limitation in breakdown voltage roadmaps (up to 1200 V) in power devices in vertical electron paths. Compatible with conventional GaN growth platforms, as the substrates are thermally matched to GaN, it offers low defect density, high crystal quality, and low wafer bow. As previous sections described, the high quality GaN power devices enabled are potential for a higher switching speed, simpler and smaller form, and higher-temperature operation.

The Naval Research Laboratory (NRL), Kyma and Qromis Technology reported some material characterization studies. It’s interesting that GaN device layers up to 15μm were demonstrated with a wafer bow of 1 μm for growth on 150-mm-diameter substrates [186]. The development to manufacture 200 mm freestanding GaN from 300 mm QST^®^ [187] also looks promising. According to several reports, the thermal conductivity and CTE relationship for different substrates are depicted in Figure 22. It also shows a high quality AlGaN/GaN buffers grown on substrates with a less mismatch in coefficient of thermal expansion (CTE). Figure 23 shows an illustration of evaluating the vertical buffer leakage currents in both reverse and forward bias mode, and it exhibits a maximum of reverse current of 1 μA/mm at 25 °C and 10 μA/mm at 150 °C as the reverse voltage exceeds 700 V. The leakage current increases by ~3 orders from 25 to 150 °C [188]. According to a MIT group, a GaN vertical power FinFETs on engineered substrate was demonstrated [189]. Figure 24 shows the schematic of the quasi-vertical device architecture, which consists of 132 fins with 100 nm, 700 nm and 21 um widths, spacing and length respectively. It exhibits a current density of J_DS_=3.8 kA/cm^2^ at V_GS_= 1.5 V and V_DS_= 4 V, and a maximum g_m_ = 2 kS/cm^2^ at V_DS_ = 4 V. The current density in each fin is higher than 30 kA/cm^2^ at the same bias condition. They also benchmark vertical and quasi-vertical MOSFETs on non-GaN substrates, as shown in Figure 25. 

## 5. Heterogeneous Integration of GaN HEMT

The conventional silicon-based RF devices fabricated on 32 nm Si complementary metal-oxide-semiconductor (CMOS) with a cut-off frequency of 445 GHz were exhibited and the cut-off frequency are expected to further scale [190]. However, despite having impressive cut-off frequency, Si CMOS is not well suited with high voltage or high-power density due to lower breakdown voltage whereas GaN devices are more suitable for these types of applications. Hence, co-integrating Si with GaN on a single chip may help in achieving high power and high-performance application. The main motivation for integrating GaN and CMOS is due to the superior GaN performance in fast power switching and the high functionality of CMOS logic, reduction in interconnect distance as well as losses, smaller form factor, reduction in power consumption, lower cost, and lower assembling complexity [191,192]. There are two types of GaN and CMOS integration variants on wafer-level namely Monolithic Integration and Heterogeneous Integration (HI). In the past few years, the key technology development and production is heterogeneous integration. For high-frequency applications in space and defense and the 5G application in the commercial world, heterogeneous integration for RF has become an essential task. Typically, hetero-integration of RF devices is done with different semiconductor materials not only CMOS to address the required specific performance like Diverse Accessible Heterogeneous Integration (DAHI) technology [193]. HI methods can be wafer to wafer (WTW), chip to wafer (CTW), or Chip to Chip (CTC), etc [194]. CTW and CTC are usually used for integrating dissimilar heterogeneous materials as it minimizes coefficient of Thermal Expansion (CTE) as well as wafer warpage issues which enable known good die (KGD) and pretty good die (PGD) that helps in achieving good yield. In WTW integration, CTE issues are challenging, and the yield is compromised. High frequency 3D HI faces challenges such as implementing effective 3D screening method as in some cases chiplets don’t have test pads and there is insufficient availability of process design kit (PDK) with RF functionality, co-simulation capability, and 3D parasitic extraction [194]. One heterogeneous integration method that makes the GaN power system compatible with CMOS fabrication using SOI substrate was stated by the IBM research division [195] as shown in Figure 26.

A DC-DC boost converter was designed using GaN power transistors integrated with bipolar-CMOS-DMOS (BCD) which combines both the advantages of high-voltage low-loss GaN devices and high-integration BCD circuits [196]. The designed GaN2BCD technology is a promising power converter application platform. Deeply scaled E/D-mode GaN HEMTs integrated with monolithically integrable GaN Schottky diodes were able to offer advantages in MMIC applications [142]. GaN-on-Si monolithic microwave integrated circuits (MMICs) were fabricated on 200-mm-diameter using a fully CMOS-compatible fabrication process which enables integration of wafer-level 3D GaN MMICs with Si CMOS circuits for performance and functionality enhancement while the size, weight, power, and cost is reduced [197]. As shown in Figure 27, a team from Raytheon [198] in the United States successfully demonstrated fabricating GaN HEMTs in windows on SOI wafers containing Si CMOS transistors, with DC and RF performance comparable to GaN HEMTs on SiC substrate, as well as a first GaN–Si CMOS heterogeneously integrated MMIC: GaN amplifier with CMOS gate bias control circuitry (a current mirror) and heterogeneous interconnects, as shown in Figure 28.

For heterogeneous integration, wafer bonding is one of the most promising integration approach for integrating group III-V materials and CMOS on Si [191]. Monolithic structure can be done by direct wafer bonding [199] or heteroepitaxy [200]. Direct wafer bonding can be used for integrating non-lattice matched semiconductors and also for integrating different crystal structures. Moreover, no additional intermediate layers are required and can also integrate two or more wafers. However, it requires a very flat, smooth, and particle-free surface along with fitting wafer diameter and chip sizes. 

A new 3D integrated circuit (3DIC) solution, System on Integrated Chips (SoICTM), was developed by Taiwan Semiconductor Manufacturing Company (TSMC) [201] to integrate active and passive chips into a new integrated SoC system. Comparing the typical 3DIC stacking with SoIC, the latter offers higher I/O density bonding density, lower energy consumption/ bit data, lower electrical parasites, and lower thermal resistance [190] which might help in unleashing the boundary of IC designer on heterogeneous integrations in future 5G, AI, mobile, and HPC applications. IMEC developed NaNO-TSV (Through Silicon Vias) connection for heterogeneous integration as 3D system-on-chip (3D-SoC) integration technology which possess a wafer-to-wafer bonding approach combined with via-last TSV connection [202]. Finally, Wafer-level packaging (WLP) of the heterogeneous integrated devices is required to be protected from the environment [203]. WLP also eliminates assembly equipment, reduces package cost, and minimizes the chip size as well as provides high yield and high reliability.

## 6. Conclusions

In the previous GaN HEMTs development roadmap, the heterogeneous epitaxy has been one of the issues affecting devices performance. The wide application of compound semiconductors has attracted wide attention and become matured gradually, including and ranging from RF power amplifiers to electronic systems. The demand tendency for power devices is increasing, especially in electric vehicles and the fast-charging applications. As high-frequency communications keep developing, the GaN HEMT technology will be very critical. Traditional silicon substrates will no longer be only GaN HEMTs template but will be replaced by other smooth and friendly substrate for some applications. These advanced substrate technologies efficiently improve device characteristics, performance, and reliability. It will bring thicker GaN buffer layer, high thermal conductivity, and high resistance substrate in the future high-power high-frequency components. In addition, the heterogeneous integration of GaN HEMTs and CMOS structure have become a new direction. In this article, we provide a brief and comprehensive overview of these important technology developments.

## Figures and Tables

**Figure 1 micromachines-12-01159-f001:**
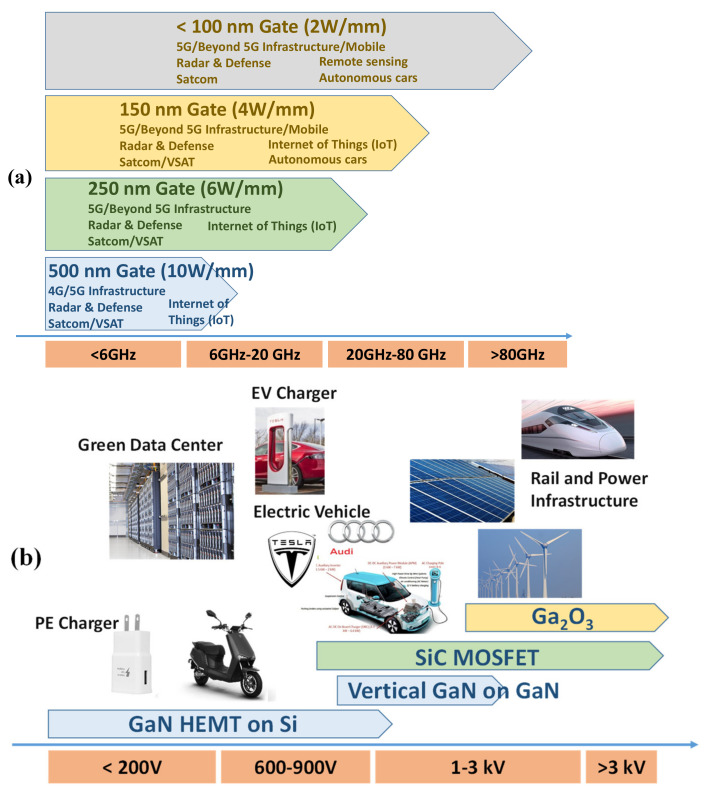
(**a**) A roadmap of RF GaN HEMTs technology. (**b**) A roadmap of Power GaN HEMTs technology.

**Figure 2 micromachines-12-01159-f002:**
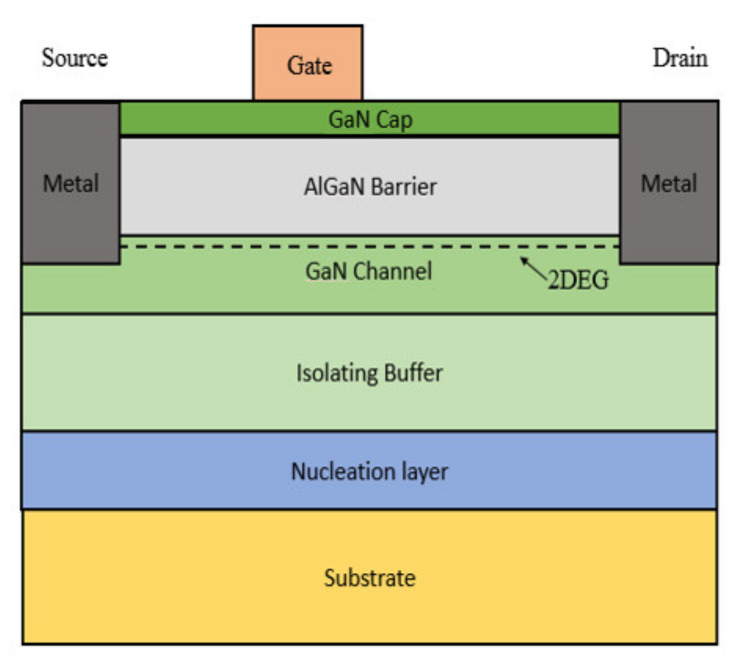
The basic structure of GaN HEMT.

**Figure 3 micromachines-12-01159-f003:**
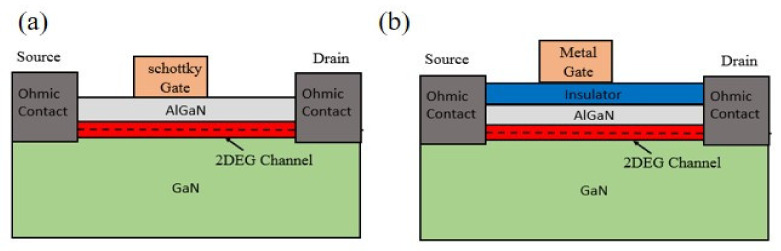
(**a**) Schottky gate D-mode HEMT (**b**) Insulated gate D-mode HEMT.

**Figure 4 micromachines-12-01159-f004:**
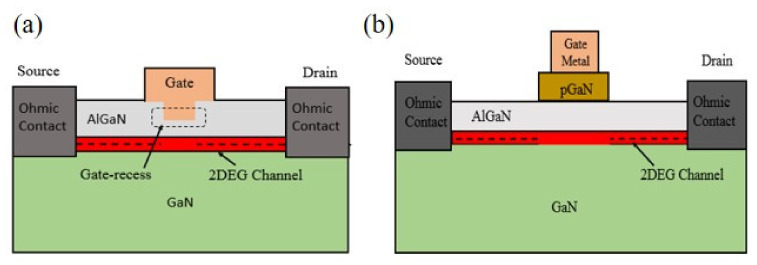
(**a**) Basic structure of Gate-recess HEMT. (**b**) Schematic structure of p-GaN Gate HEMT.

**Figure 5 micromachines-12-01159-f005:**
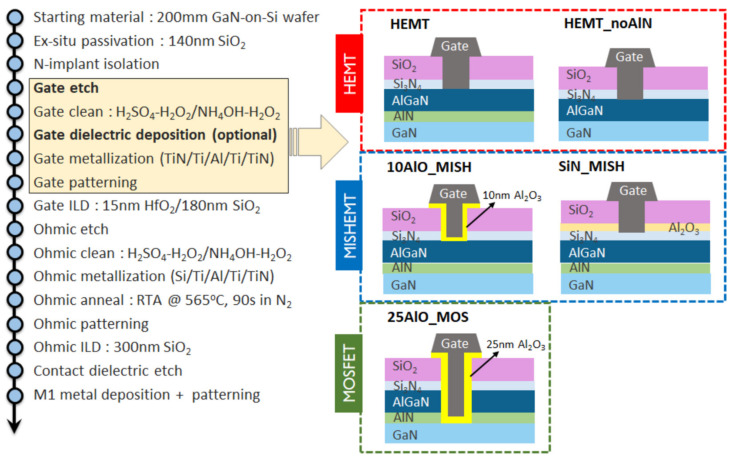
Gate-first process flow for the fabrication of GaN devices. Five device splits were realized based on differences in gate processing [98].

**Figure 6 micromachines-12-01159-f006:**
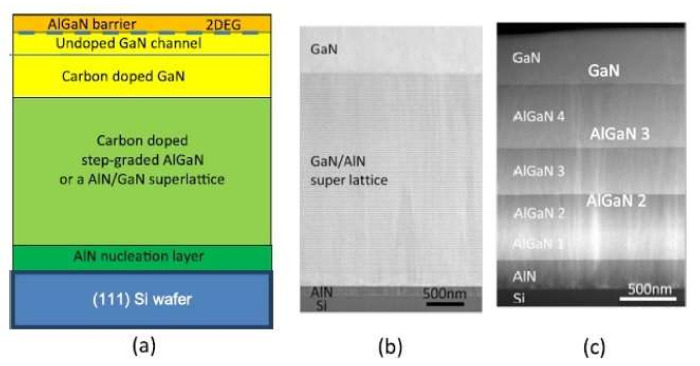
(**a**) Schematic cross-section of the typical epitaxial layer structure used for the manufacture of GaN-on-Si HEMTs. (**b**) TEM image of a GaN/AlN superlattice buffer layer and (**c**) a step graded AlGaN buffer layer, both on Si substrates [12].

**Figure 7 micromachines-12-01159-f007:**
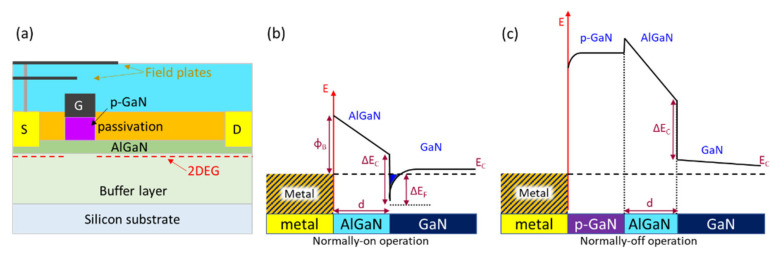
(**a**) Cross-sectional schematic of p-GaN gate HEMT [135] and (**b**) schematic of the operation principle of the normally on HEMT and (**c**) normally off HEMT [74].

**Figure 8 micromachines-12-01159-f008:**
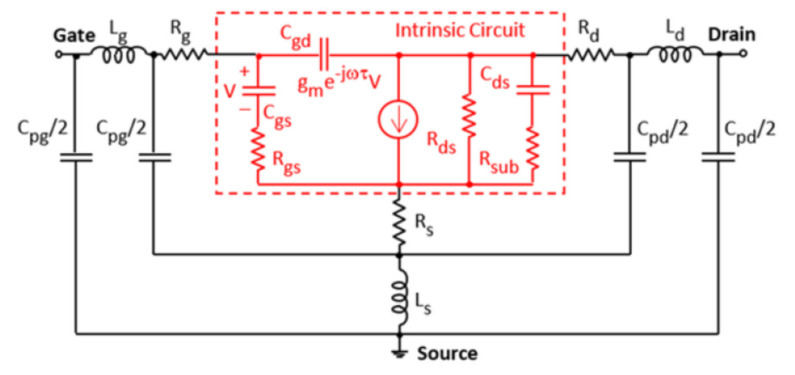
Small-signal equivalent circuit for the tested MOSFETs.

**Figure 9 micromachines-12-01159-f009:**
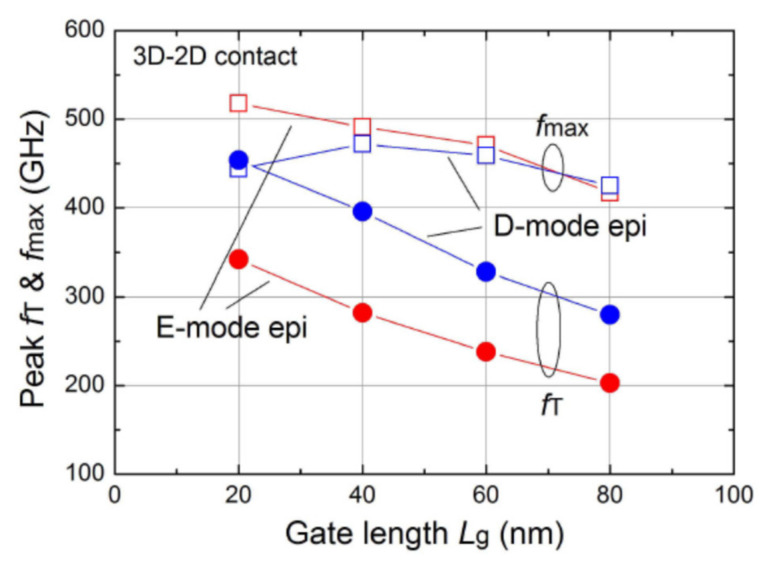
F_T_ and F_MAX_ versus Lg [142].

**Figure 10 micromachines-12-01159-f010:**
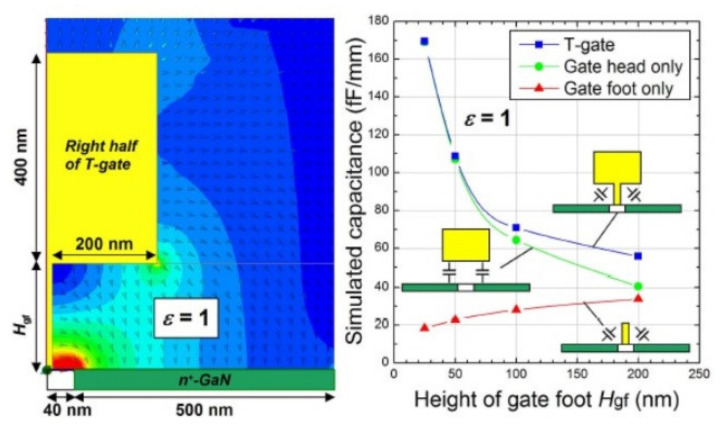
Simulation for T-gate capacitance [142].

**Figure 11 micromachines-12-01159-f011:**
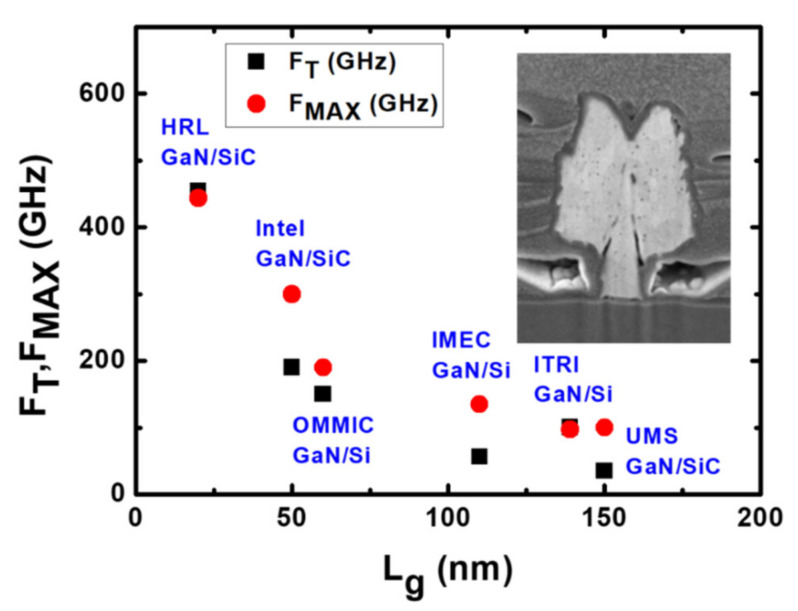
Benchmark for frequency versus Lg.

**Figure 12 micromachines-12-01159-f012:**
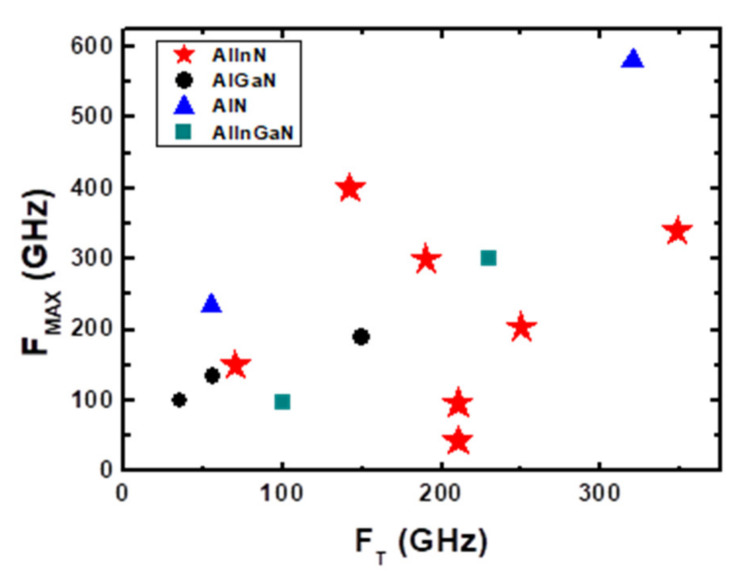
Comparison of the measured f_T_ and f_MAX_ in GaN-based HEMTs from literature.

**Figure 13 micromachines-12-01159-f013:**
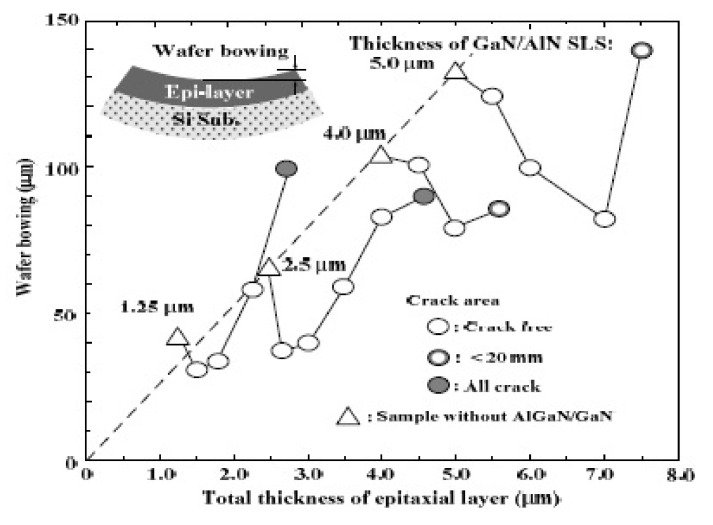
Wafer bowing as a function of total epitaxial layer thickness [163].

**Figure 14 micromachines-12-01159-f014:**
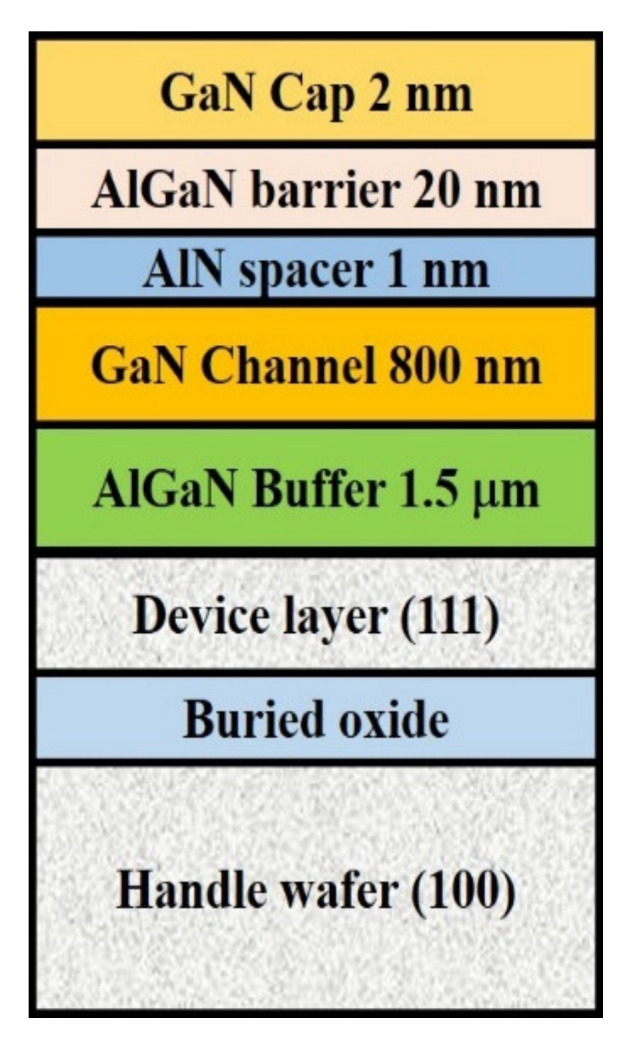
Schematic cross-section of AlGaN/GaN HEMT on SOI substrate.

**Figure 15 micromachines-12-01159-f015:**
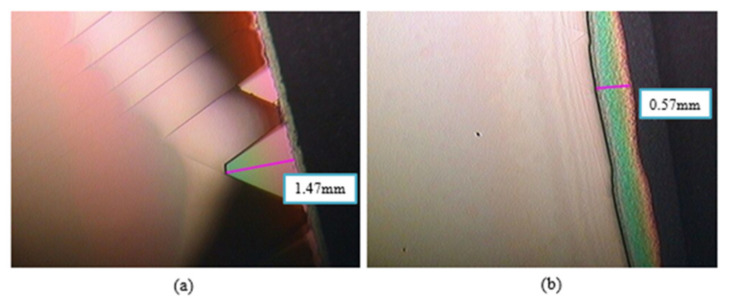
Top section of OM image (**a**) Sample A. (**b**) Sample B.

**Figure 16 micromachines-12-01159-f016:**
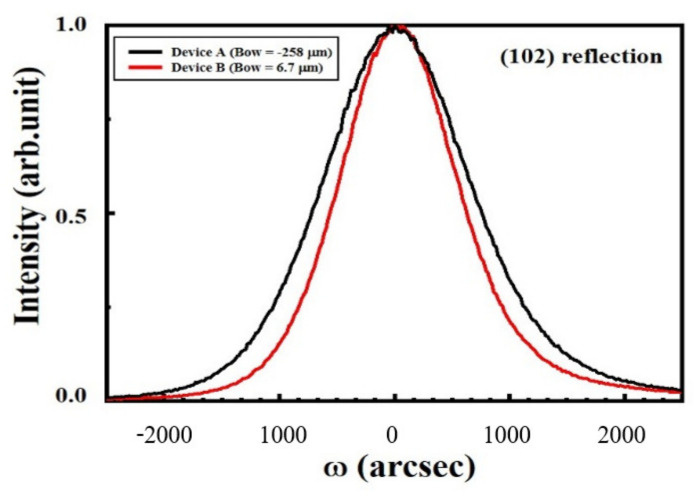
Asymmetric (102) XRD ω-scan rocking curves of substrate with big BOW value (black) and small BOW value (red) measured from surface.

**Figure 17 micromachines-12-01159-f017:**
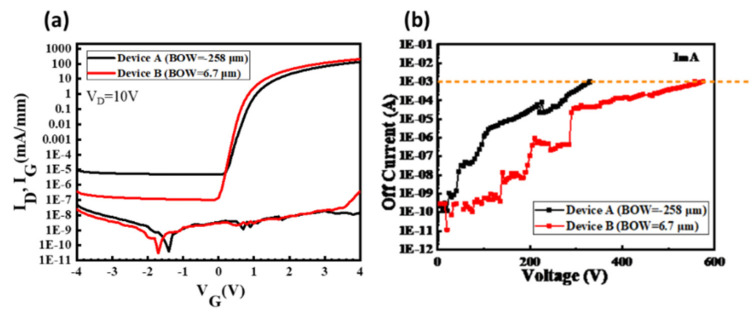
(**a**) I_DS_-V_GS_ characteristics of HEMTs with big BOW value (Device A, black line) and small BOW value (Device B, red line) (V_DS_ = 10 V). (**b**) Three-terminal off-state characteristic of the E-mode HEMTs at V_GS_ =−5 V.

**Figure 18 micromachines-12-01159-f018:**
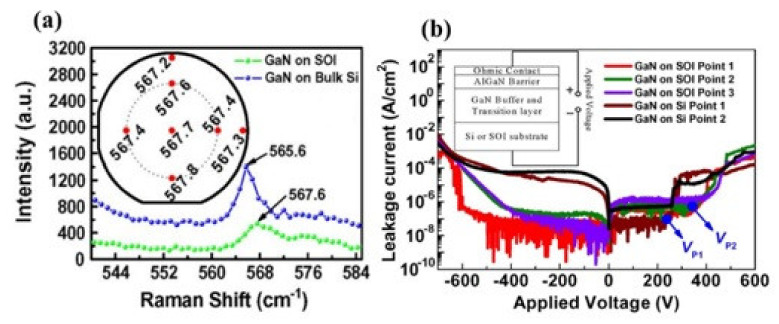
(**a**) GaN-on-SOI and GaN-on-Si (bulk) wafers under Micro-Raman spectroscopy. The E2 peak on the GaN-on-SOI wafer was mapped (Inset) (**b**) Characteristics of vertical leakage on GaN-on-SOI and GaN-on-Si (bulk) platforms [169].

**Figure 19 micromachines-12-01159-f019:**
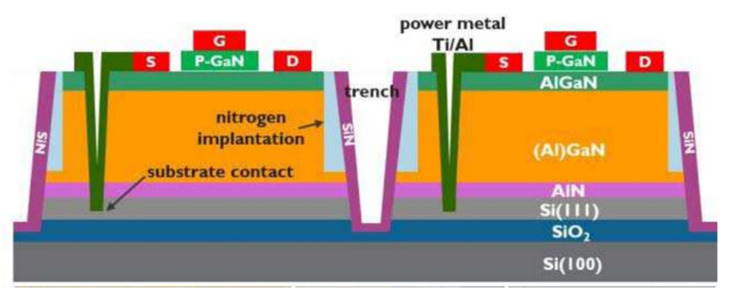
Schematic cross-section of the isolated e-mode p-GaN HEMT [177].

**Figure 20 micromachines-12-01159-f020:**
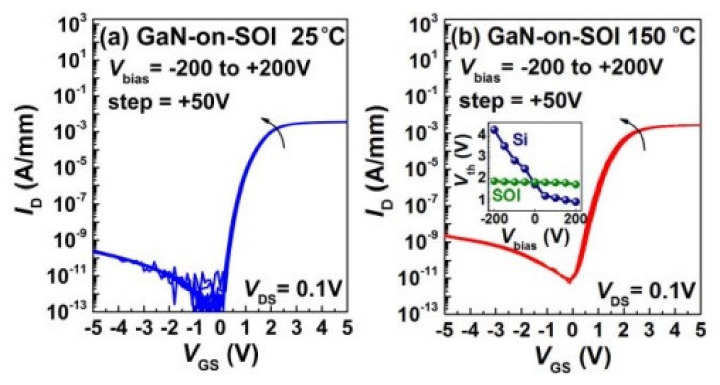
Transfer characteristics while simultaneously biasing the silicon substrate from −200 V to 200 V (**a**) with a common silicon substrate (**b**) with SOI substrate [177].

**Figure 21 micromachines-12-01159-f021:**
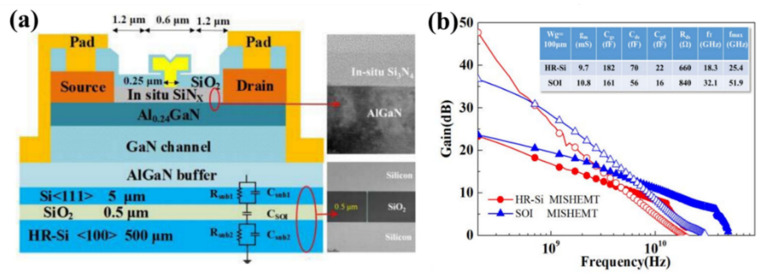
(**a**) Cross-sectional structure of MISHEMT on SOI substrate (**b**) high frequency parameters for AlGaN/GaN MISHEMT on SOI substrate(blue) and AlGaN/GaN MISHEMT on HR-Si substrate (red). [185].

**Figure 22 micromachines-12-01159-f022:**
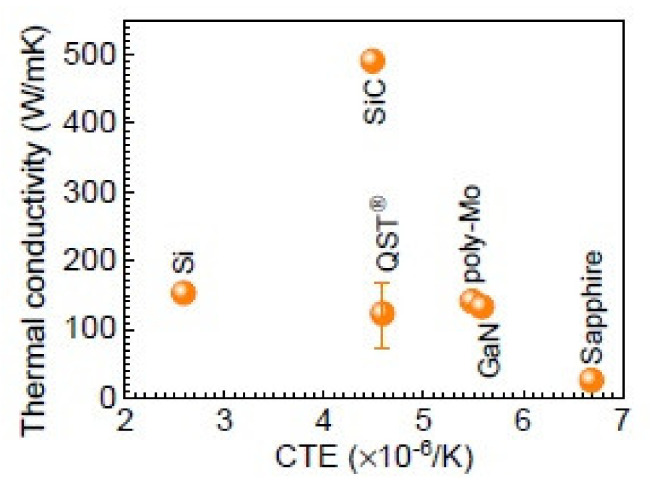
Different substrate for coefficient of thermal expansion (CTE) and thermal conductivity [188].

**Figure 23 micromachines-12-01159-f023:**
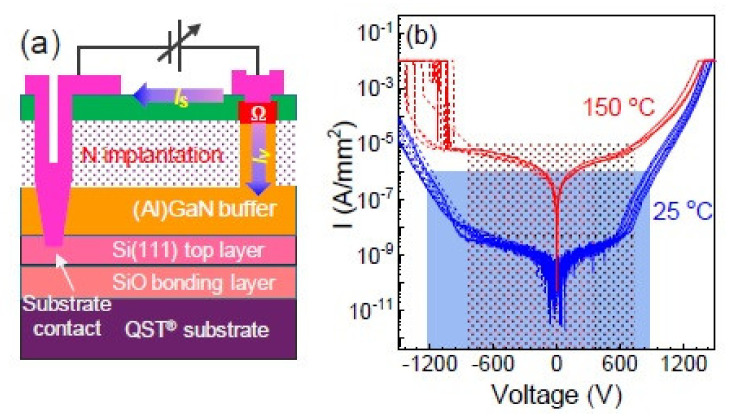
(**a**) The illustration of vertical buffer leakage measurements (**b**) The leakage current density under voltage bias for a 5.6 mm-thick buffer grown on 200 mm GaN-on-QST^®^ substrate [188].

**Figure 24 micromachines-12-01159-f024:**
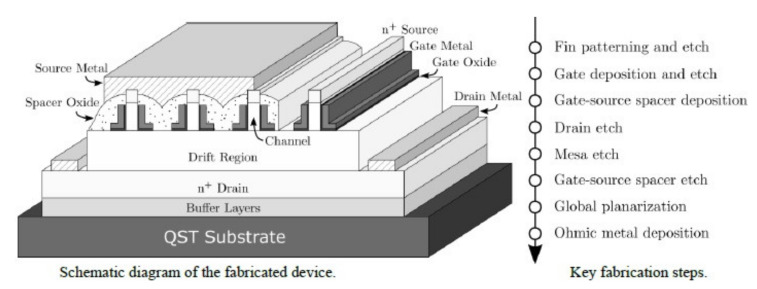
A schematic diagram of the vertical GaN FinFET devices and fabrication steps [189].

**Figure 25 micromachines-12-01159-f025:**
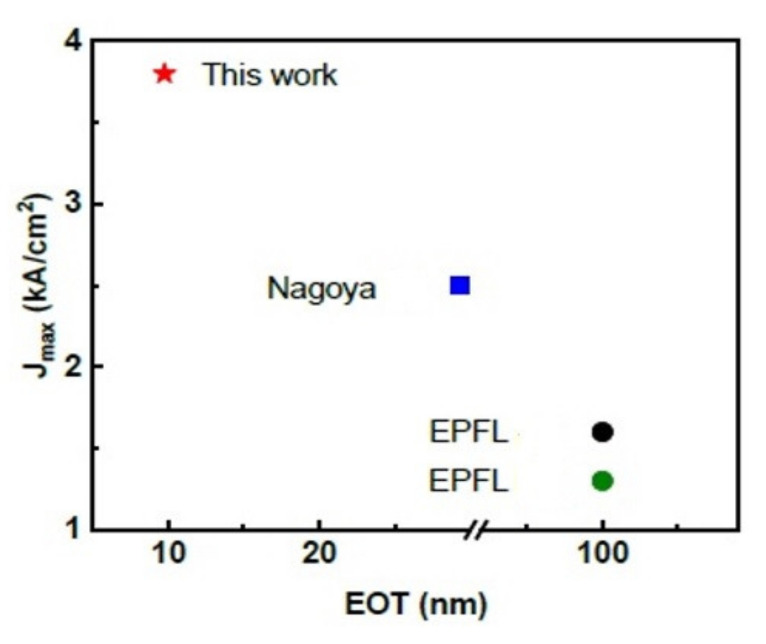
Benchmarking maximum current density in vertical GaN-on-Silicon transistors as a function of equivalent oxide thickness [189].

**Figure 26 micromachines-12-01159-f026:**
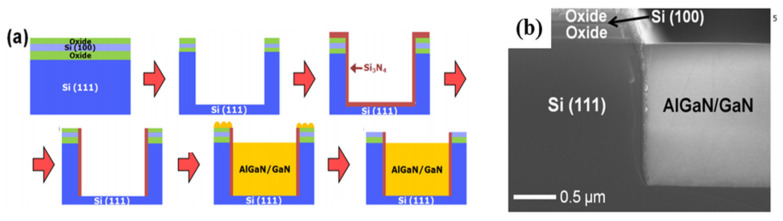
(**a**) Heterogeneous integration of SOI substrate with top Si (100) and bottom Si (111) and GaN epitaxial (**b**) SEM cross-section [195].

**Figure 27 micromachines-12-01159-f027:**
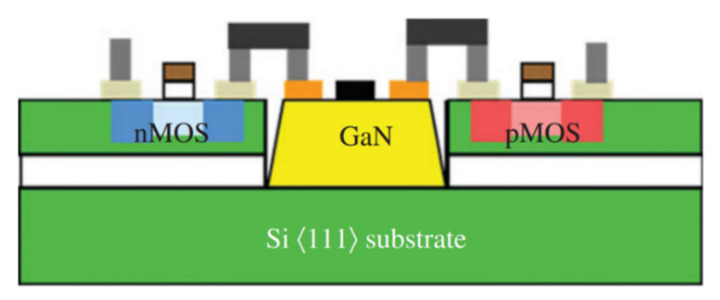
GaN HEMT and Si CMOS are heterogeneously combined on a modified SOI wafer [198].

**Figure 28 micromachines-12-01159-f028:**
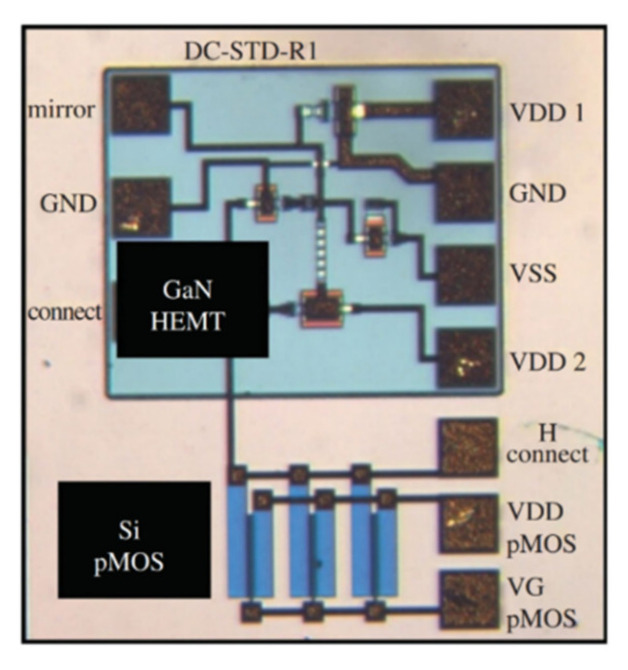
GaN–Si CMOS heterogeneously integrated MMIC [198].

**Table 1 micromachines-12-01159-t001:** Temperature-dependent thermal conductivity of GaN/different substrate materials, TBR values for GaN/substrate interfaces used in simulations are listed [41,44].

Material	Thermal Conductivity κ (W/m-K)	$\mathrm{TBR}\;(m^{2}K/\mathrm{GW})$	Thermal Expansion Coefficient, (10^−6^/K)	Lattice Mismatch with GaN (%)
GaN	160 (${300/T)}^{1.4}$	–	–	–
Sapphire	35 (${300/T)}^{1}$	10–40	39	16
Si	150 (${300/T)}^{1.3}$	10–40	54	17
SiC	420 (${300/T)}^{1.3}$	30–60	3.2	4
Diamond	1200 (${300/T)}^{1}$	20–50	62.5	12

**Table 2 micromachines-12-01159-t002:** The lattice and thermal mismatch of Si, SiC, Sapphire, AlN, and GaN.

Mismatch	Si	SiC	Sapphire	AlN	GaN
Crystal Structure	FCC	HCP	HCP	HCP	HCP
Lattice Constant (Å)	5.43	3.08	4.758	3.112	3.189
Lattice Mismatch (%)	−16.9	3.5	16.08	2.4	–
Thermal Expansion (10^−6^ K)	3.59	4.3	7.3	4.15	5.59
Thermal Mismatch (%)	55	30	−23	34	–

**Table 3 micromachines-12-01159-t003:** Device characteristic of different Boron doped level in handle wafer.

Sample	A	B
Dope-level	Light doped	Heavy doped
Doping concentration (atoms-cm^−3^)	1.35 × 10^15^~1.49 × 10^16^	5.95 × 10^19^~1.26 × 10^20^
Bowing (μm)	−258	6.7
FWHM (arcsec) (1 0 2)	1484	1188

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
