# Peer review of "Development of GaN HEMTs Fabricated on Silicon, Silicon-on-Insulator, and Engineered Substrates and the Heterogeneous Integration"

_micromachines, 2021, doi:10.3390/mi12101159_

Round 1

Reviewer 1 Report

In this paper the authors, made a successful attempt to present a comprehensive review of GaN HEMT technology on different substrates both for power electronic and RF applications. Overall it was a good read. However, in my opinion the authors should address the following comments before the paper is accepted for publication into this journal.  

a)  N-polar GaN HEMT devices show state of the art performances in terms of power density and PAE at 94 GHz. As such, it is worthwhile to talk about the benefit of these devices in a paragraph or two. Refer to the following papers for more details:

[1] W.Liu et al “6.2 W/m and Record 33.8% PAE at 94 GHz From N-Polar GaN Deep Recess MIS- HEMTs With ALD Ru Gates” IEEE MWCL 31, 748 (2021)

[2] A. Arias et al., "High performance N-polar GaN HEMTs with OIP3/Pdc ∼12dB at 10GHz," 2017 IEEE Compound Semiconductor Integrated Circuit Symposium (CSICS), 2017, pp. 1-3, doi: 10.1109/CSICS.2017.8240456.

b) Another area of active research is the development of ultra linear GaN RF transistors. Authors should elaborate on how the linearity of GaN HEMT can be improved at high frequency (>30 GHz) in a paragraph or two. 

c) Another active area of development is the GaN based CMOS Technology led by MIT and Cornell university in collaboration with Intel. Therefore, this review paper should dedicate some space to talk about the development of such technology which is useful for both power electronics and RF applications. Also it might be worthwhile to cite the following papers in this regard:

[1] N. Chowdhury, Q. Xie, M. Yuan, K. Cheng, H. W. Then, and T. Palacios, “Regrowth-free GaN-based complementary logic on a Si substrate,” IEEE Electron Device Letters 41, 820–823 (2020).

[2] K. Nomoto et al., "GaN/AlN p-channel HFETs with Imax >420 mA/mm and ~20 GHz fT / fMAX," 2020 IEEE International Electron Devices Meeting (IEDM), 2020, pp. 8.3.1-8.3.4, doi: 10.1109/IEDM13553.2020.9371994.

[3] N. Chowdhury, J. Lemettinen, Q. Xie, Y. Zhang, N. S. Rajput, P. Xiang, K. Cheng, S. Suihkonen, H. W. Then, and T. Palacios, “p-channel GaN transistor based on p-GaN/AlGaN/GaN on Si,” IEEE Electron Device Letters 40, 1036–1039 (2019).

Author Response

Dear reviewer: 

Thanks a lot for your suggestions, and the related reply as below, 

a) A good information was provided, and N-polar GaN HEMT devices of the art performances in terms of power density and PAE at 94 GHz. We have a supplements (N-polar vs Ga-polar) that explain information in the introduction parts.

 b) We have the intensive and related information to explain in section 2.3_RF HEMTs on Si substrate.

 c) The new stage and supplied sentence for GaN based CMOS Technology were into the introduction parts_"1.5 GaN-based CMOS technology".

Reviewer 2 Report

Hsu et al. in their review article entitled "Development of GaN HEMTs fabricated on Silicon, Silicon-on-insulator, and engineered substrates and the heterogeneous integration"
presented valuable information.
Minor revision suggested with the following comments: If possible, the author should include electron mobility, 2DEG density, and sheet resistance information as a function of surface to channel distance.
As this is a review, so author should create a section "Future Directions" at the end of the draft. 

Author Response

Dear reviewer:

Thanks a lot for your suggestions and the related reply as below,

1)  2DEG functions of surface to channel distance has been formed by the analysis in simulations, in several group as attached to files. Additional, we have a supplement of (section_1.3) Ga-polar vs N-polar to explain 2DEG related to barrier/channel face domination, and section_2.2 power HEMT and section_2.3 RF HEMT to explain different designs in 2DEG system. 

2) The conclusion with future direction has been added in the end of draft.
